# 2α-Substituted Vitamin D Derivatives Effectively Enhance the Osteoblast Differentiation of Dedifferentiated Fat Cells

**DOI:** 10.3390/biom14060706

**Published:** 2024-06-15

**Authors:** Michiyasu Ishizawa, Masashi Takano, Atsushi Kittaka, Taro Matsumoto, Makoto Makishima

**Affiliations:** 1Division of Biochemistry, Department of Biomedical Sciences, Nihon University School of Medicine, Itabashi-ku, Tokyo 173-8610, Japan; 2Faculty of Pharmaceutical Sciences, Teikyo University, 2-11-1 Kaga, Itabashi-ku, Tokyo 173-8605, Japan; mtakano@pharm.teikyo-u.ac.jp (M.T.); akittaka@pharm.teikyo-u.ac.jp (A.K.); 3Division of Cell Regeneration and Transplantation, Department of Functional Morphology, Nihon University School of Medicine, Tokyo 173-8610, Japan; matsumoto.taro@nihon-u.ac.jp

**Keywords:** vitamin D derivatives, vitamin D receptor, osteoblast differentiation, dedifferentiated fat cells

## Abstract

The active form of vitamin D_3_, 1α,25-dihydroxyvitamin D_3_ [1,25(OH)_2_D_3_], is a principal regulator of calcium homeostasis through activation of the vitamin D receptor (VDR). Previous studies have shown that 2α-(3-hydroxypropyl)-1,25D_3_ (O1C3) and 2α-(3-hydroxypropoxy)-1,25D_3_ (O2C3), vitamin D derivatives resistant to inactivation enzymes, can activate VDR, induce leukemic cell differentiation, and increase blood calcium levels in rats more effectively than 1,25(OH)_2_D_3_. In this study, to further investigate the usefulness of 2α-substituted vitamin D derivatives, we examined the effects of O2C3, O1C3, and their derivatives on VDR activity in cells and mouse tissues and on osteoblast differentiation of dedifferentiated fat (DFAT) cells, a cell type with potential therapeutic application in regenerative medicine. In cell culture experiments using kidney-derived HEK293 cells, intestinal mucosa-derived CaCO_2_ cells, and osteoblast-derived MG63 cells, and in mouse experiments, O2C2, O2C3, O1C3, and O1C4 had a weaker effect than or equivalent effect to 1,25(OH)_2_D_3_ in VDR transactivation and induction of the VDR target gene *CYP24A1*, but they enhanced osteoblast differentiation in DFAT cells equally to or more effectively than 1,25(OH)_2_D_3_. In long-term treatment with the compound without the medium change (7 days), the derivatives enhanced osteoblast differentiation more effectively than 1,25(OH)_2_D_3_. O2C3 and O1C3 were more stable than 1,25(OH)_2_D_3_ in DFAT cell culture. These results indicate that 2α-substituted vitamin D derivatives, such as inactivation-resistant O2C3 and O1C3, are more effective than 1,25(OH)_2_D_3_ in osteoblast differentiation of DFAT cells, suggesting potential roles in regenerative medicine with DFAT cells and other multipotent cells.

## 1. Introduction

1α,25-dihydroxyvitamin D_3_ [1,25(OH)_2_D_3_] is a required molecule for the maintenance of calcium and bone homeostasis, and its function is mediated by the vitamin D receptor (VDR), a member of the nuclear receptor superfamily [1]. The binding of 1,25(OH)_2_D_3_ induces the dynamic interaction of VDR with its heterodimeric partner, the retinoid X receptor (RXR), and coactivators such as steroid receptor coactivator 1 (SRC1) [2]. VDR regulates the expression of genes involved in calcium metabolism and vitamin D metabolism, such as *CYP24A1* and *CYP3A4* [2,3]. VDR is also involved in other biological and physiological processes, such as cell differentiation and inflammatory responses [1]. Pharmacologic VDR activation is expected to be useful in the treatment of cancer and inflammatory and autoimmune diseases as well as bone and mineral disorders. Although alfacalcidol [4] and eldecalcitol [5] have been successfully applied for the treatment of osteoporosis, the clinical application of vitamin D derivatives for diseases other than bone and mineral disorders has been limited by adverse effects, particularly hypercalcemia [6].

Modification of the side chain of vitamin D is thought to control stability by altering the mode of binding to VDR and its interaction with transcriptional cofactors and by controlling the substrate specificity of vitamin D-metabolizing enzymes, thereby exhibiting tissue and action selectivity [7]. We previously developed vitamin D derivatives modified at the 2α position to form a hydrogen bond with Arg274 of the ligand-binding domain of VDR [8,9]. Among them, 2α-(3-hydroxypropyl)-1,25D_3_ (O1C3) and 2α-(3-hydroxypropoxy)-1,25D_3_ (O2C3) (Figure 1) are superior to 1,25(OH)_2_D_3_ in their ability to activate VDR in vitro, to induce leukemia cell HL60 differentiation, and to raise blood calcium in rats [10,11,12,13]. O1C3 and O2C3 are resistant to metabolism by the vitamin D-inactivating enzymes, such as CYP24A1 and CYP3A4 [8,14]. The resistance of vitamin D derivatives to inactivation may prolong the biological effects in target cells. 

Dedifferentiated fat (DFAT) cells are multipotent cells obtained by the ceiling culture of mature adipocytes [15,16]. Since DFAT cells show very similar characteristics to adipose-derived stem/stromal cells and can differentiate into multiple lineages such as adipocytes, osteoblasts, and chondrocytes [16], they have potential for application in regenerative medicine and injury healing [17,18]. 1,25(OH)_2_D_3_ stimulates osteogenic differentiation of human adipose-derived stromal cells and mesenchymal stem cells (MSCs) [19,20,21]. We previously showed that 1,25(OH)_2_D_3_ and its derivatives enhance osteoblast differentiation of DFAT cells [22]. In this study, we investigated the effect of vitamin D derivatives modified at the 2α-position on VDR target gene expression in human cells and mouse tissues and osteoblast differentiation in DFAT cells to evaluate their biological activities for potential biological application.

## 2. Materials and Methods

### 2.1. Vitamin D Derivatives

The synthesis and chemical properties of vitamin D derivatives modified at the 2α-position (O2C2, O2C3, O2C4, O1C1, O1C2, O1C3, and O1C4) were reported previously [10,11,12] (Figure 1). O2C3 [2α-(3-hydroxypropoxyl)-1α,25(OH)_2_D_3_], which is the C2-epimer of ED-71, and its derivatives were synthesized from D-glucose [10], and O1C3 [2α-(3-hydroxypropyl)-1α,25(OH)_2_D_3_] was synthesized from D-xylose [11]. O2C3 and O1C3 have 3 times and 1.8 times higher VDR binding affinities, respectively, than 1,25(OH)_2_D_3_ [10,11]. 1,25(OH)_2_D_3_ was purchased from Fujifilm Wako Pure Chemical Corporation (Osaka, Japan). 1,25(OH)_2_D_3_ and vitamin D derivatives were dissolved in ethanol.

### 2.2. Cell Cultures

Human kidney-derived HEK293 cells (RIKEN Cell Bank, Tsukuba, Japan) were cultured in Dulbecco’s modified Eagle’s medium (DMEM) containing 5% fetal bovine serum (FBS), 100 U/mL penicillin, and 0.1 mg/mL streptomycin (Nacalai Tesque, Kyoto, Japan). Human colon carcinoma CaCO_2_ cells and human osteosarcoma MG63 cells (RIKEN Cell Bank) were cultured in minimum essential medium containing 10% FBS, 100 U/mL penicillin, and 0.1 mg/mL streptomycin. Cell cultures were maintained at 37 °C in a humidified atmosphere containing 5% CO_2_.

### 2.3. Luciferase Reporter Assays

The expression vectors pCMX-VDR, pCMX-VP16-VDR, pCMX-GAL4-RXRα, and pCMX-GAL4-SRC-1, in which the nuclear receptor-interacting domains of SRC-1 (amino acids 595−771; GenBank accession code U90661) were inserted into the pCMX-GAL4 vector, and the luciferase reporter vectors VDR-responsive Sppx3-tk-LUC and GAL4- responsive MH100(UAS) × 4-tk-LUC reporter vectors were used for luciferase reporter assays [22]. HEK293 cells were transfected with 50 ng of Spp × 3-tk-LUC reporter plasmid, 10 ng of pCMX-β-galactosidase, 15 ng of pCMX-VDR for each well of a 96-well plate using the calcium phosphate coprecipitation method [22]. Eight hours after transfection, cells were treated with 1,25(OH)_2_D_3_ or vitamin D derivative at 0.1, 1, or 10 nM in the culture medium for 16–24 h and were analyzed for luciferase and β-galactosidase activity using a luminometer and a microplate reader, respectively (Molecular Devices, Sunnyvale, CA, USA). A mammalian two-hybrid assay for VDR-RXR or VDR-SRC1 interaction used 50 ng of MH100(UAS) × 4-tk-LUC reporter plasmid, 10 ng of pCMX-β-galactosidase, 15 ng of pCMX-GAL4-RXRα or pCMX-GAL4-SRC1, and 15 ng of pCMX-VP16-VDR for each well of a 96-well plate. Luciferase data were normalized to the internal β-galactosidase activity.

### 2.4. Mouse Experiments

Eight-week-old male C57BL/6J mice (CLEA Japan, Tokyo, Japan) were raised on a standard diet (CE-2; CLEA Japan) in a specific pathogen-free facility and maintained under a controlled temperature (23 ± 1 °C) and humidity (45–65%) with free access to water. Each compound (1,25(OH)_2_D_3_, O2C2, O2C3, or O1C3) (12.5 nmol/kg, diluted in phosphate-buffered saline) was injected intraperitoneally to mice. Six hours after injection, mice were euthanized with carbon dioxide, and blood and tissue samples were collected. To avoid the influence of collection time differences, we repeated sample collections from each group (for example, sample collection was repeated in the order of control mouse, 1,25(OH)_2_D_3_ mouse, O1C3 mouse in the experiments to compare mice in each group). Collected samples were frozen on dry ice and stored until analysis. The plasma calcium and phosphorus concentrations were quantified with a Calcium C Test wako and Phospho C Test wako (Fujifilm Wako) [23,24]. All animal experiments were performed according to protocols that adhered to the Nihon University Animal Care and Use Committee and conformed to the ARRIVE guidelines.

### 2.5. Human DFAT Cell Isolation and Culture

Samples of human subcutaneous adipose tissue were obtained from patients undergoing surgery in the Department of Pediatric Surgery, Nihon University Itabashi Hospital (Tokyo, Japan). The patients gave written informed consent, and the Ethics Committee of Nihon University School of Medicine approved the study. Preparation of DFAT cells using the ceiling culture was performed according to the method described by Matsumoto et al. [16]. Briefly, small pieces (approximately 1 g) of the adipose tissue were digested with 0.1% type I collagenase solution (Koken Co., Ltd., Tokyo, Japan) and were centrifuged after filtration. Then, floating adipocytes were collected, washed with phosphate-buffered saline, and plated in 25 cm^2^ culture flasks (Thermo Fisher Scientific, Waltham, MA, USA) that were filled completely with CSTI-303MSC (Cell Science & Technology Institute, Miyagi, Japan) containing 20% FBS with 5 × 10^4^ cells per flask. The adipocytes immediately floated up and subsequently adhered to the top ceiling surface of the flask within 2–3 days of the culture. The adhered cells lost their lipid droplets and changed their morphology to fibroblast-like cells. On day 7, the flasks were inverted after media removal, and cells were cultured in 5 mL CSTI-303MSC containing 20% FBS. The medium was changed every 4 days until cells reached confluence. Cells were passaged by standard methods of trypsinization and were used for experiments as DFAT cells at passage 2–4. The expression rates of surface markers on the DFAT cells used in this study are shown in Table 1, consistent with the MSC-like properties of DFAT cells [16,25].

### 2.6. Reverse Transcription and Quantitative Real-Time PCR Analysis

For gene expression analysis for cells, HEK293, CaCO_2_, and MG63 cells, (1 × 10^4^ per well) were plated in a 24-well plate, or DFAT cells (1.2 × 10^5^ per well) were plated in a 12-well plate. After 24 h, cells were replaced with DMEM containing 10% FBS and 10 nM of each compound (1,25(OH)_2_D_3_, O2C2, O2C3, O2C4, O1C1, O1C2, O1C3, or O1C4) or ethanol control and cultured for 24 h. Mouse tissue samples were also used for mRNA expression analysis.

Total RNAs from cells or tissue samples were prepared by the acid guanidine thiocyanate phenol/chloroform method [23]. cDNAs were synthesized using the ImProm-II reverse transcription system (Promega, Madison, WI, USA) [26]. Real-time PCR was performed on the StepOne Plus Real-time PCR System (Thermo Fisher Scientific, Norristown, PA, USA) using Power SYBR Green PCR master mix (Thermo Fisher Scientific). Primer sequences are shown in Table 2. mRNA levels were normalized to the level of *GAPDH* mRNA or 18S rRNA and calculated relative to those in the 1,25(OH)_2_D_3_ treatment to compare the effects of vitamin D derivatives to those of 1,25(OH)_2_D_3_.

### 2.7. Differentiation Assay

For osteogenic differentiation, confluent cells were incubated for a week in osteogenic medium (OM; DMEM containing 10% FBS, 100 nM dexamethasone (Merck KGaA, Darmstadt, Germany), 10 mM β-glycerophosphate (Merck KGaA), and 50 mM L-ascorbic acid-2-phosphate (Merck KGaA)) with 10 nM of each compound (1,25(OH)_2_D_3_, O2C2, O2C3, O2C4, O1C1, O1C2, O1C3, or O1C4). The OM (with a test compound) was changed on day 4 or left unchanged. The level of alkaline phosphatase (ALP) activity was determined on day 7. After washing with phosphate-buffered saline, cells were suspended in deionized distilled H_2_O with 0.05% TritonX-100 and freeze-thawed. ALP activity in the supernatant was determined at 405 nm with a lab assay ALP kit (Fujifilm Wako). Total protein content was determined with a BCA protein assay kit (Thermo Fisher Scientific) and was used for normalization of ALP levels.

### 2.8. High-Performance Liquid Chromatography

For estimation of the stability of compounds, human DFAT cells plated in 12-well plates were cultured in OM plus 1 μM 1,25(OH)_2_D_3_, O2C3, or O1C3 for 3 days. After culture, compounds were extracted using three volumes of chloroform/methanol (3:1, *v*/*v*) from cells and medium. For control experiments, each compound was incubated in OM without cells for 3 days and extracted by the same method. The organic phase was recovered and dried under reduced pressure, and the residue was dissolved in acetonitrile. High-performance liquid chromatography (HPLC) was performed under the following conditions: column, InertSustain C18 (5 μm; 4.6 mm × 250 mm) (GL Sciences Inc., Tokyo, Japan); UV detection, 265 nm; flow rate, 1.0 mL/min; column temperature, 40 °C; mobile phase, linear gradient of 20−100% acetonitrile/water per 25 min followed by 100% acetonitrile for 20 min [27].

### 2.9. Statistical Analysis

Data are presented as means ± S.D. We performed one-way ANOVA followed by Dunnett’s multiple comparisons or unpaired Student’s *t* test to assess significant differences using Prism 8 (version 8.4.2; GraphPad Software, La Jolla, CA, USA).

## 3. Results

### 3.1. Effects of Vitamin D Derivatives on VDR Transactivation Activity and Interaction of VDR with RXR and SRC-1

At first, we examined the effect of O2C3 and its derivatives on VDR activity. O2C2 and O2C3 induced VDR transactivation activity slightly less than 1,25(OH)_2_D_3_ (Figure 2A). The half maximal effective concentrations (EC_50_s) for VDR transactivation of 1,25(OH)_2_D_3_, O2C2, O2C3, and O2C4 were 0.005 nM, 0.03 nM, 0.05 nM, and 0.66 nM, respectively. These compounds also induced the interaction of VDR with RXRα and the coactivator SRC-1 less strongly than 1,25(OH)_2_D_3_ (Figure 2B,C). The EC_50_s for VDR-RXR interaction by 1,25(OH)_2_D_3_, O2C2, O2C3 and O2C4 were 0.003 nM, 0.02 nM, 0.02 nM, and 0.05 nM, respectively, and those for VDR-SRC-1 interaction were 0.04 nM, 0.2 nM, 0.2 nM, and 9.6 nM, respectively, consistent with their VDR transactivation activities. Their VDR effects were stronger on the order of 1,25(OH)_2_D_3_ > O2C2 = O2C3 > O2C4.

We also examined the effect of O1C3 and its derivatives on VDR activity. The EC_50_s for VDR transactivation of 1,25(OH)_2_D_3_, O1C1, O1C2, O1C3, and O1C4 were 0.005 nM, 0.7 nM, 0.04 nM, 0.01 nM, and 0.03 nM, respectively (Figure 2D). Although the potency of O1C4 on VDR transactivation was lower, the efficacy of O1C4 was slightly higher than that of the 1,25(OH)_2_D_3_. These compounds induced weaker interactions of VDR with RXRα and SRC-1 than 1,25(OH)_2_D_3_ (Figure 2E,F). Since the EC_50_s for VDR-RXR interaction by O1C1, O1C2, O1C3, and O1C4 were 0.3 nM, 0.08 nM, 0.05 nM, and 0.04 nM, respectively, and those for VDR-SRC-1 interaction were >3 μM, 0.7 nM, 0.2 nM, and 0.4 nM, respectively, their VDR effects were stronger on the order of 1,25(OH)_2_D_3_ > O1C3 = O1C4 > O1C2 > O1C1.

The effect of vitamin D derivatives on the expression of the endogenous VDR target gene *CYP24A1* was examined next. Because 1,25(OH)_2_D_3_ induces the concentration-dependent expression of *CYP24A1* in intestinal mucosa-derived SW480 cells in the range of 1 nM to 100 nM [23], and we observed similar effects of 1,25(OH)_2_D_3_ in human kidney-derived HEK293 cells in preliminary experiments, we compared the effects of vitamin D derivatives to those of 1,25(OH)_2_D_3_ at 10 nM. We treated HEK293 cells, intestinal mucosa-derived CaCO_2_ cells, and osteoblast-derived MG63 cells with 1,25(OH)_2_D_3_ or vitamin D derivative at 10 nM for 24 h. O2C2 induced *CYP24A1* expression but at a lower level than 1,25(OH)_2_D_3_, while the effect of O2C3 or O2C4 was not significant (Figure 3A). O1C3 induced *CYP24A1* expression less effectively than 1,25(OH)_2_D_3_, and O1C1, O1C2, and O1C4 were not effective (Figure 3B). In CaCO_2_ cells, O2C2, O2C3 induced *CYP24A1* expression but less effectively than 1,25(OH)_2_D_3_ (Figure 3C). While O1C2 was less effective, O1C4 increased *CYP24A1* expression as well as 1,25(OH)_2_D_3_, and O1C3 increased the expression at a higher level than 1,25(OH)_2_D_3_ (Figure 3D). In MG63 cells, O2C2 and O2C3 increased *CYP24A1* expression to similar levels by 1,25(OH)_2_D_3_, while O2C4 was less effective (Figure 3E). O1C2 induced *CYP24A1* expression to the same extent as 1,25(OH)_2_D_3_, and O1C3 and O1C4 increased its expression at higher levels than 1,25(OH)_2_D_3_ (Figure 3F). O2C2, O2C3, O1C2, O1C3, O1C4 also increased the expression of *BGLAP*, which is a VDR target and encodes osteocalcin, a marker of osteoblast differentiation [2,28], to similar levels as 1,25(OH)_2_D_3_ (Figure 3G,H). Interestingly, O2C4 and O1C1 were effective in *BGLAP* induction, although they were less effective than 1,25(OH)_2_D_3_. Thus, MG63 cells are more sensitive to O2C2 and O2C3 than HEK293 cells and CaCO_2_ cells for VDR target gene induction, suggesting a cell-selective action of the 2α-substituted vitamin D derivatives.

### 3.2. Effects of Vitamin D Derivatives on Tissue Cyp24a1 Expression and Plasma Calcium and Phosphorus Levels in Mice

Next, we examined in vivo effects of vitamin D derivatives. As reported previously [29], intraperitoneal injection of 1,25(OH)_2_D_3_ induced *Cyp24a1* expression in the kidney, duodenum, jejunum, and ileum in mice (Figure 4A). Interestingly, O2C2 and O2C3 were more effective than 1,25(OH)_2_D_3_ in mice. Serum calcium levels were reported to gradually increase up to 24 h after intraperitoneal injection of 1,25(OH)_2_D_3_ [30]. We observed a significant increase in plasma calcium levels 6 h after 1,25(OH)_2_D_3_ injection (Figure 4B). O2C3 slightly increased plasma calcium levels as well as 1,25(OH)_2_D_3_, while the effect of O2C2 was not significant (Figure 4B). O1C3 increased *Cyp24a1* mRNA levels in the kidney, duodenum, jejunum, and ileum more effectively than 1,25(OH)_2_D_3_ but did not increase plasma calcium levels at this time point (Figure 4C,D). Treatment of mice with 1,25(OH)_2_D_3_ increased serum phosphorus levels [31]. O2C2, O2C3, and O1C3, like 1,25(OH)_2_D_3_, increased plasma phosphorus levels (Figure 4B,D). These results indicate that the vitamin D derivatives O2C2, O2C3, and O1C3 exert VDR action to a comparable or more effective level than 1,25(OH)_2_D_3_ in vivo, although their effects are weaker in cultured cells. 

### 3.3. Effects of Vitamin D Derivatives on Osteoblast Differentiation in Human DFAT Cells

We examined the effect of vitamin D derivatives on the VDR target gene *CYP24A1*’s expression in human DFAT cells. O2C2, O2C3, O1C2, and O1C4 increased *CYP24A1* expression but more weakly than 1,25(OH)_2_D_3_, and O2C4 and O1C1 were not effective (Figure 5A). Interestingly, O1C3 induced *CYP24A1* expression more effectively than 1,25(OH)_2_D_3_. *SPP1* and *BGLAP* are VDR target genes related to osteogenesis [2]. O2C2, O1C2, O1C3, and O1C4 increased *SPP1* expression, and the effect of O1C3 was stronger than 1,25(OH)_2_D_3_ (Figure 5B). While 1,25(OH)_2_D_3_ was not effective in *BGLAP* induction, O1C2 and O1C3 increased *BGLAP* expression (Figure 5C). The expression of *RUNX2*, a gene related to osteogenesis, or *PPARG*, an adipocyte marker gene, was not altered by any treatment (Figure 5D,E). 

We previously showed that 1,25(OH)_2_D_3_ and its derivatives enhance osteoblast differentiation of DFAT cells [22]. We examined the effect of the 2α-substituted vitamin D derivatives on ALP activity, a marker of osteoblast differentiation, in human DFAT cells. We induced osteoblast differentiation with OM in the absence or presence of test compounds. The media were changed on day 4 with fresh OM plus test compounds, and ALP activity in cells was analyzed on day 7. Interestingly, all derivatives except O1C1 increased ALP activity in cells cultured with OM (Figure 6A). Of note, O2C3, O1C3, and O1C4 enhanced ALP activity more effectively than 1,25(OH)_2_D_3_, although their VDR transactivation activities were weaker than 1,25(OH)_2_D_3_ (Figure 2). Next, we treated cells with vitamin D derivatives without the medium exchange. Under this experimental condition, O2C2, O2C3, O1C2, O1C3, and O1C4 enhanced ALP activity more effectively than 1,25(OH)_2_D_3_, with activity values of 172%, 177%, 182%, 365%, and 299% compared to 1,25(OH)_2_D_3_, respectively (Figure 6B). These results indicate that 2α-substituted vitamin D derivatives, such as O1C3, are more effective in enhancing osteoblast differentiation of DFAT cells than 1,25(OH)_2_D_3_, specifically in the condition without the medium change.

Finally, we evaluated the stability of compounds in cell culture. 1,25(OH)_2_D_3_, O2C3, and O1C3 were incubated in OM without or with DFAT cells for 3 days, and the stability was evaluated with HPLC. 1,25(OH)_2_D_3_ with DFAT cells with OM decreased by about 68% compared to the cell-free condition (Figure 7). O2C3 and O1C3 decreased by only 33% and 20%, respectively. These results show that O2C3 and O1C3 are more stable than 1,25(OH)_2_D_3_ in cell culture.

## 4. Discussion

In this study, we show that 2α-hydroxyalkoxylated or 2α-hydroxyalkylated vitamin D derivatives are more effective in the induction of VDR target gene expression in vivo and in the enhancement of osteoblast differentiation in DFAT cells than 1,25(OH)_2_D_3_, although these VDR ligands have weaker transactivation activity than 1,25(OH)_2_D_3_.

Hydroxyalkoxylation or hydroxyalkylation of the 2α-position enhances the binding affinity of vitamin D derivatives O2C3 and O1C3 to the VDR by 1.8- and 3-fold, respectively, compared to 1,25(OH)_2_D_3_ [12]. X-ray crystallographic analysis of these compounds with VDR shows that the binding mode to VDR is similar to that of previously developed VDR agonists, and that there is maintenance of the position and structure of the activation function 2 domain and the conformation of hydrogen bonds of 1-OH to Ser237 and Arg274, 3-OH to Tyr143 and Ser278, 25-OH to His305 and His397, which are particularly important for binding to the ligand-binding domain of VDR [32]. The difference in the binding mode of 1,25(OH)_2_D_3_ and the derivatives is that the Phe150 and 2α substituents of the VDR form a weak van der Waals contact, and that at the site where three water molecules mediate binding to the VDR in 1,25(OH)_2_D_3_, two water molecules are excluded in O1C3 and O2C3, forming a direct hydrogen bond. It is thought that this part of the molecule enhances the binding affinity of the VDR by affecting the interaction with Arg274, which is important for hydrogen bonding with 1-OH in 1,25(OH)_2_D_3_.

Contrary to the enhanced binding affinity for VDR, short-term treatment with O2C3 or O1C3 showed weaker transcriptional activation of VDR than 1,25(OH)_2_D_3_. O2C3 and O1C3 had a weak ability to interact with SRC-1 and activate endogenous VDR in HEK293 cells (Figure 2). These results are consistent with the previous report by Takahashi et al. [12]. In contrast, O2C3 and O1C3 activated endogenous VDR target gene expression in MG63 cells as effectively as or more effectively than 1,25(OH)_2_D_3_ (Figure 3). SRC-1 is highly expressed in HEK293 cells compared to MG63 cells [26]. These results suggest that cofactors other than SRC-1 are involved in cell-selective VDR activity in MG63 cells.

In vitro metabolic experiments by Yasuda et al. reveal that O2C3 is not readily metabolized by CYP24A1 and that O1C3 is not readily metabolized by CYP24A1 and CYP3A4 [14]. They also mention that O1C3 is more metabolically resistant than ED-71, although this is a preliminary study. O2C3, O1C3, and their derivatives (O2C2, O1C2, and O1C4) exhibited stronger activity in osteogenic differentiation of human DFAT cells than 1,25(OH)_2_D_3_ in the condition without the medium change (Figure 6). O2C3 and O1C3 were more stable than 1,25(OH)_2_D_3_ in DFAT cell culture (Figure 7). These findings indicate that 2α-substituted vitamin D derivatives such as O2C3 and O1C3 are inactivation-resistant and can stably exhibit biological activity in cells.

O2C2, O2C3, and O1C3 induced *Cyp24a1* expression in VDR target tissues in mice as effectively as more effectively than 1,25(OH)_2_D_3_ (Figure 4). The in vivo effects of the vitamin D derivatives may be influenced by their resistance to inactivating enzymes. O2C2 and O1C3 increased plasma phosphorus levels but did not affect calcium levels, whereas O2C2 increased both phosphorus and calcium levels as well as 1,25(OH)_2_D_3_. Pharmacological properties, such as cofactor recruitment, and pharmacokinetics, including tissue-selective accumulation and stability, may influence the selective functions of vitamin D derivatives in mice. 

Our results in Figure 6 suggest that the local injection of transplantable multipotent cells together with the compound could be used for regenerative medicine. DFAT cells can be prepared more quickly and at a lower cost than induced pluripotent stem cells, although these cells are not omnipotent [16]. DFAT cells have MSC-like properties, such as immunophenotypes (CD73^+^, CD90^+^, CD105^+^, CD31^−^, CD45^−^, and HLA-DR^−^) (Table 1) and tri-lineage (adipogenic, osteogenic, and chondrogenic) differentiation potential [16,33,34], consistent with the minimal criteria for defining MSCs [25]. DFAT cells can also re-differentiate into skeletal muscle cells [35], smooth muscle cells [36], cardiomyocytes [37], and endothelial cells [38]. Furthermore, DFAT cells re-differentiate into osteoblasts more efficiently than adipose stromal cells [39]. Implantation of DFAT cells promotes periodontal regeneration and healing in animal models of periodontitis [40]. The use of 2α-substituted vitamin D derivatives, such as O1C3, may be helpful in regenerative medicine and cell therapy.

## 5. Conclusions

2α-Hydroxyalkoxylated or 2α-hydroxyalkylated vitamin D derivatives, such as O2C3 and O1C3, are effective in inducing VDR target gene expression in vivo, and in enhancing osteoblast differentiation in DFAT cells. They may have potential use as VDR-targeting drugs in regenerative medicine and cell therapy.

## Figures and Tables

**Figure 1 biomolecules-14-00706-f001:**
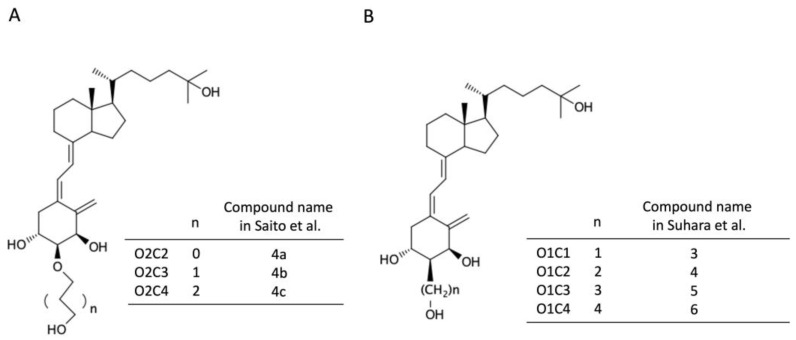
Chemical structures of 2α-substituted vitamin D derivatives. (**A**) 2α-Hydroxyalkoxylated derivatives, O2C2, O2C3, and O2C4. (**B**) 2α-Hydroxyalkylated derivatives, O1C1, O1C2, O1C3, and O1C4. The details about the compounds were reported previously [10,11].

**Figure 2 biomolecules-14-00706-f002:**
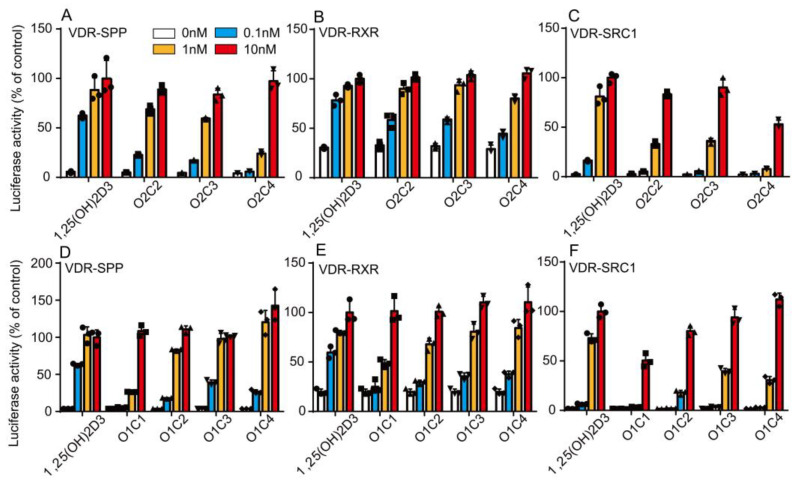
Effects of 1,25(OH)_2_D_3_ and vitamin D derivatives on VDR transactivation activity and interactions of VDR with RXR and SRC-1. The effects of O2C2, O2C3, and O2C4 on VDR transactivation (**A**), VDR–RXR interaction (**B**), and VDR–SRC1 interaction (**C**), and those of O1C1, O1C2, O1C3, and O1C4 on VDR transactivation (**D**), VDR–RXR interaction (**E**), and VDR–SRC1 interaction were compared to those of 1,25(OH)_2_D_3_. HEK293 cells were transfected with pCMX−VDR and TK-Spp × 3-LUC reporter plasmid, CMX-VP16-VDR, CMX-GAL4-RXRα, and MH100(UAS) × 4-tk-LUC reporter plasmid, and CMX-VP16-VDR, CMX-GAL4-SRC-1, and MH100(UAS) × 4-tk-LUC reporter plasmid for VDR transactivation activity (**A**,**D**), interaction of VDR with RXRα (**B**,**E**), and that with SRC-1 (**C**,**F**), respectively. Cells were treated with a range of concentrations of each compound (0–10 nM). Luciferase activity values are expressed relative to those of 10 nM 1,25(OH)_2_D_3_, which are set at 100%.

**Figure 3 biomolecules-14-00706-f003:**
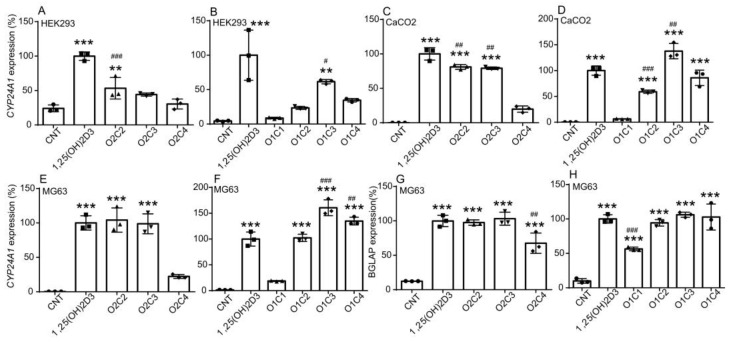
Effects of 1,25(OH)_2_D_3_ and vitamin D derivatives on mRNA expression of the VDR target genes in human cells. The effects of O2C2, O2C3, and O2C4 (**A**,**C**,**E**,**G**) and those of O1C1, O1C2, O1C3, and O1C4 (**B**,**D**,**F**,**H**) were compared to those of 1,25(OH)_2_D_3_. Human kidney-derived HEK293 cells (**A**,**B**), intestinal mucosa-derived CaCO_2_ cells (**C**,**D**), and osteoblast-derived MG63 cells (**E**–**H**) were treated with a vehicle (ethanol) control (CNT), 1,25(OH)_2_D_3_ or vitamin D derivative (10 nM) for 24 h, and mRNA expression of *CYP24A1* (**A**–**F**) and *BGLAP* (**G**,**H**) was determined with reverse transcription and quantitative real-time PCR analysis. mRNA levels were normalized to the level of 18S rRNA and expressed relative to those of cells treated with 1,25(OH)_2_D_3_, which are set at 100%. One-way ANOVA followed by Dunnett’s multiple comparisons. ** *p* < 0.01, *** *p* < 0.001 versus CNT; # *p* < 0.05, ## *p* < 0.01, ### *p* < 0.001 versus 1,25(OH)_2_D_3_.

**Figure 4 biomolecules-14-00706-f004:**
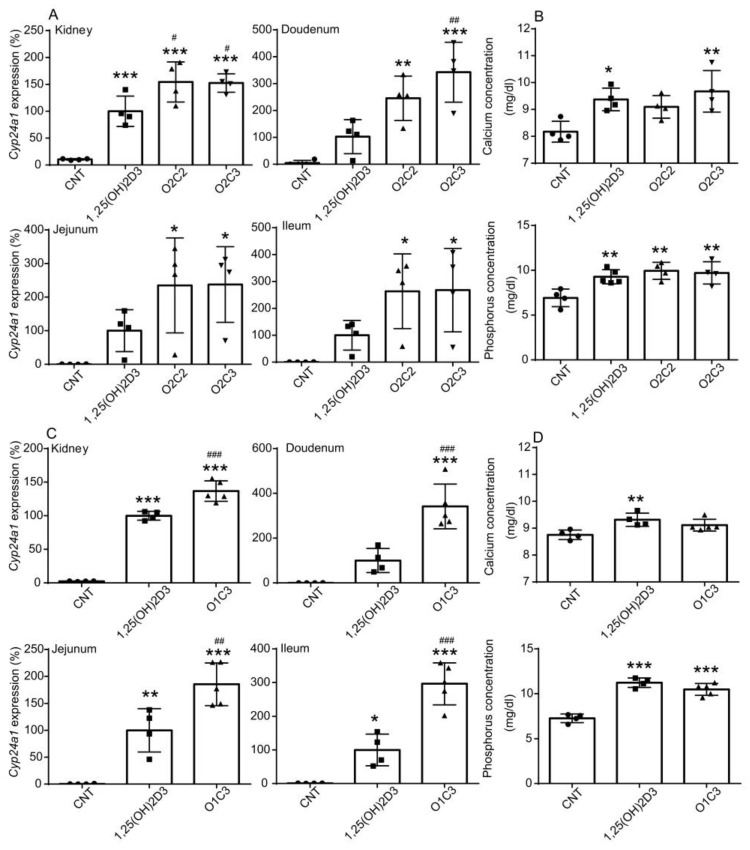
Effects of vitamin D derivatives on *Cyp24a1* expression in the kidney and small intestine, and plasma calcium and phosphorus levels in mice. (**A**) Effects of O2C2 and O2C3 on *Cyp24a1* mRNA expression in the kidney, duodenum, jejunum, and ileum were compared to those of 1,25(OH)_2_D_3_. (**B**) Effects of O2C2 and O2C3 on plasma calcium and phosphorus levels were compared to those of 1,25(OH)_2_D_3_. Effects of O1C3 on *Cyp24a1* mRNA expression (**C**) and plasma calcium and phosphorus levels (**D**) were also examined. Mice were administered vehicle (ethanol) control (CNT), 12.5 nmol/kg 1,25(OH)_2_D_3_, O2C2, O2C3, or O1C3 via intraperitoneal injection, and blood and tissue samples were collected 6 h after injection. mRNA levels were normalized to the level of *GAPDH* mRNA and expressed relative to those of 1,25(OH)_2_D_3_-treated mice, which are set at 100%. Data are presented as means ± S.D. One-way ANOVA followed by Dunnett’s multiple comparisons. * *p* < 0.05, ** *p* < 0.01, *** *p* < 0.001 versus CNT; # *p* < 0.05; ## *p* < 0.01, ### *p* < 0.001 versus 1,25(OH)_2_D_3_.

**Figure 5 biomolecules-14-00706-f005:**
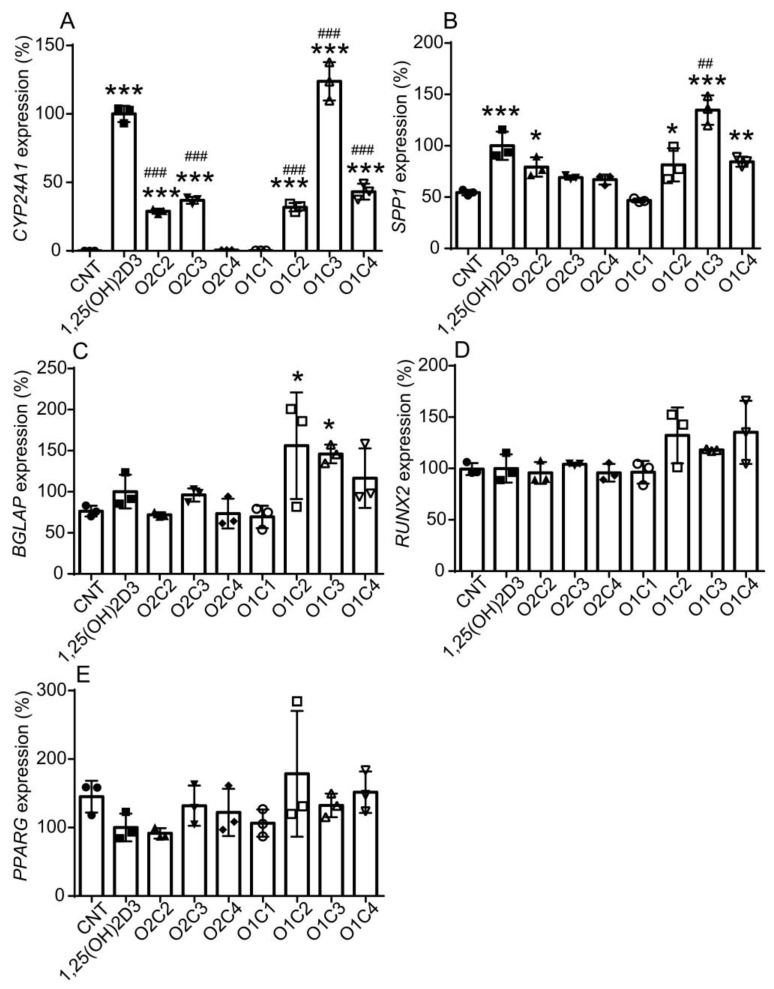
Effects of vitamin D derivatives on the expression of the VDR target gene *CYP24A1* (**A**), the osteoblast marker genes *SPP1* (**B**), *BGLAP* (**C**), and *RUNX2* (**D**), and the adipocyte marker gene *PPARG* (**E**) in human DFAT cells. Cells were treated with vehicle (ethanol) control (CNT), 10 nM 1,25(OH)_2_D_3_, or vitamin D derivative for 24 h, and the expression of each gene was determined via reverse transcription and quantitative real-time PCR analysis. mRNA levels were normalized to the level of 18S rRNA and expressed relative to those of cells treated with 1,25(OH)_2_D_3_, which are set at 100%. One-way ANOVA followed by Dunnett’s multiple comparisons. * *p* < 0.05, ** *p* < 0.01, *** *p* < 0.001 versus CNT; ## *p* < 0.01, ### *p* < 0.001 versus 1,25(OH)_2_D_3_.

**Figure 6 biomolecules-14-00706-f006:**
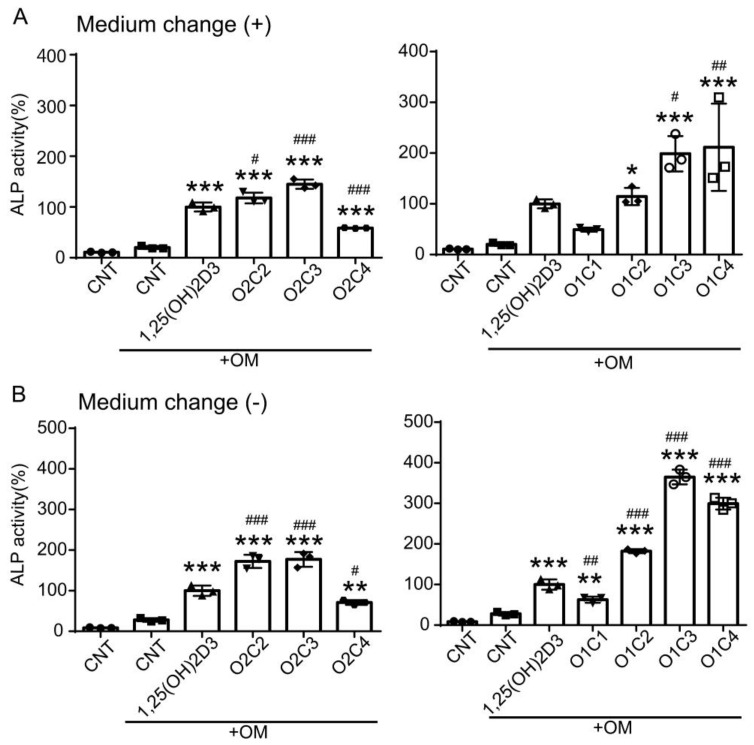
Effects of vitamin D derivatives on osteoblast differentiation in human DFAT cells. Cells were treated without or with OM in the presence of vehicle (ethanol) control (CNT), 10 nM 1,25(OH)_2_D_3_, or vitamin D derivative. OM plus test compound was changed on day 4 (**A**) or left unchanged (**B**). ALP activity was evaluated on day 7. ALP activity was determined in cell lysates and normalized to protein content and expressed relative to those of cells treated with OM plus 1,25(OH)_2_D_3_, which are set at 100%. One-way ANOVA followed by Dunnett’s multiple comparisons. * *p* < 0.05, ** *p* < 0.01, *** *p* < 0.001 versus OM + CNT (*p* = 0.07, OM + 1,25(OH)_2_D_3_ versus OM + CNT in (**A**) right panel); # *p* < 0.05, ## *p* < 0.01, ### *p* < 0.001 versus OM + 1,25(OH)_2_D_3_.

**Figure 7 biomolecules-14-00706-f007:**
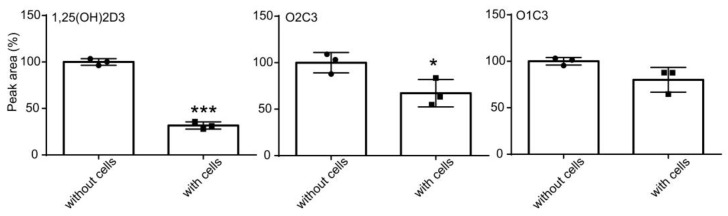
Stability of 1,25(OH)_2_D_3_, O2C3, and O1C3 in human DFAT cells. Cells were treated with OM plus 1 μM 1,25(OH)_2_D3, O2C3, or O1C3 for 3 days (“with cells”). Each compound was also incubated with OM in the absence of cells for 3 days (“without cells”). Compounds were extracted from medium +/− cells and analyzed with HPLC. Peak areas for compounds in HPLC were expressed relative to those without cells, which are set at 100%. Unpaired two-group Student’s *t* test: * *p* < 0.05, *** *p* < 0.001.

**Table 1 biomolecules-14-00706-t001:** Expression of CD105, CD90, CD73, CD31, CD45, and HLA-DR on DFAT cells.

Surface Marker	Positive Rate (%)
CD105	85.5
CD90	98.1
CD73	99.6
CD31	0.01
CD45	0.05
HLA-DR	0.10

**Table 2 biomolecules-14-00706-t002:** Primer sequences for quantitative real-time PCR analysis.

Species	Gene Symbol	Gene Name	Sequence
Human	*BGLAP*	bone gamma-carboxyglutamate protein	5′-CCA GGC GCT ACC TGT ATC AA-3′ 5′-AAG CCG ATG TGG TCA GCC AA-3′
	*CYP24A1*	cytochrome P450 family 24 subfamily A member 1	5′-TGA ACG TTG GCT TCA GGA GAA-3′ 5′-AGG GTG CCT GAG TGT AGC ATC T-3′
	*PPARG*	peroxisome proliferator activated receptor gamma	5′-CGT GGA TCT CTC CGT AAT GGA-3′ 5′-AAT AAG GTG GAG ATG CAG GCT C-3′
	*RNA18SN4*	18S rRNA	5′-GTA ACC CGT TGA ACC CCA TT-3’ 5′-CCA TCC AAT CGG TAG TAG CG-3′
	*RUNX2*	RUNX family transcription factor 2	5′-CAT TTG CAC TGG GTC ACA CGT A-3′ 5′-GAA TCT GGC CAT GTT TGT GCT C-3′
	*SPP1*	secreted phosphoprotein 1	5′-ACT CCA ATC GTC CCT ACA GT-3′ 5′-TAG ACT CAC CGC TCT TCA TG-3′
Mouse	*Cyp24a1*	cytochrome P450 family 24 subfamily A member 1	5′-TGG AGA CGA CCG CAA ACA G-3′ 5′-AGG CAG CAC GCT CTG GAT T-3′
	*Gapdh*	glyceraldehyde-3-phosphate dehydrogenase	5′-TGC ACC ACC AAC TGC TTA G-3′ 5′-GAT GCA GGG ATG ATG TTC-3′

## Data Availability

The data presented in this study are available on request from the corresponding author.

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
