# Peer review of "2α-Substituted Vitamin D Derivatives Effectively Enhance the Osteoblast Differentiation of Dedifferentiated Fat Cells"

_biomolecules, 2024, doi:10.3390/biom14060706_

Round 1

Reviewer 1 Report

Comments and Suggestions for Authors

The study from Ishizawa and coll. investigates the properties of two series of vitamin D derivatives, in terms of VDR transactivation, VDR transcriptional activity (evaluated as CYP24 expression) and VDR-induced differentiation of NFAT cells. The latter activity is particularly interesting because it could be exploited in enhancing osteoblast differentiation for regenerative medicine purposes.  

Previous studies had described the increased stability of these compounds, which were able to act as ligands of VDR but were hydroxylated by CYP24 and degraded to a lesser extent than calcitriol.

This study fully describes the properties of the O2 derivatives, whereas does not investigates the O1 derivatives in the experiments of VDR transactivation and CYP24 induction in cell lines and mice.

I would like to know the reason of this lack of results, which does not allow to reach proper and general conclusions. In fact, in discussion some sentences are not correct, due to the fact that O1 derivatives have not been fully tested.

For example, lines 291-294, and lines 336-337.

This is the main flaw of the article, which in general is interesting and the experiments are well carried out, I wonder if it is possible to add the missing data about O1 derivatives.

Some further observations:

1.       Figure 4B: Plasma calcium levels were measured in mice after 8 hours of stimulation, is this incubation time sufficient to expect a rise in calcium levels?

Figure 6: vitamin D derivatives are more effective in enhancing osteoblast differentiation of DFAT cells in the  condition without medium change. This is probably also due to their increased stability compared to calcitriol. Please comment on this in discussion. Moreover, in the experiments shown in figure 6 it would be interesting to compare the data obtained after the treatment with calcitriol with medium change and without medium change. Without medium change the ALP activity should be lower because calcitriol is partially degraded, and this would reinforce the higher stability of the derivatives as the cause or their superior performance. If you wish, you could actually express all data relative to calcitriol treatment of panel A.

Reviewer 2 Report

Comments and Suggestions for Authors

Please find my comments below.

1. Authors aimed to investigate an influence of 2α-substituted vitamin D derivatives on the osteoblast differentiation of dedifferentiated fat cells. Regarding the topic of this work, I must admit that I do not find it appropriately addressed. I recommend reconsider changing the topic.

2. Major flaws of this article are that authors did not examine the cell phenotype before and after the differentiation. Defining the cell characteristics with the use of proteomic methods (ICC, IHC, IF, flow cytometry WB or ELISA) is a must when the experiments are conducted on primary cell cultures. Thus, if the described experiment does not contain data confirming cell phenotype it is hard to judge on significance of presented results.

3. A plain description of experiment design and study groups would significantly increase a legibility of this article. Authors use interchangeably the symbols of study groups like O2C2, O2C3, O2C4 1,25D3 or a description like "vit D derivatives" or "2α-hydroxyalkoxylated/2α-hydroxyalkylated vit D", but never explained what are these compounds.

4. Authors quite often refer to their previous publications. I find it too frequent. Authors forces the potential readers to read their previous work to find an explanation of conducted experiment. Authors have cited at least 5 of their publications (~18% of all references). All the important information should be given within the present article not just be addressed in previous work.

5. It is not explained why author used 10 nM concentration of tested compounds. It should be.

6. The differentiation assay is quite superficial. The osteogenesis is a complex process which cannot be assessed only by the evaluation of ALP gene expression. This process should be investigated in more detail and supported by divergent methods, not only the qPCR.

7. I recommend exposing the used specific pairs of starters in the table to make it more comfortable for readers.

8. Data presentation require improvements. The figures are missing legends, which in my opinion would raise the legibility of presented data. The figure captions could be redacted to clearly describe what is presented in the figure. Authors should describe what were the used control groups and how the data were compared. It should be described in material and methods. For example, in the figure 2 it is unclear what are the relative light units, in the figures 3-7 it is not clear what are CNT groups. Moreover, authors mentioned that they used GAPDH or 18S rRNA as reference, but did not describe when it was applied. It is also hard to link the figure with specific assay that have been conducted by the authors.

Summarizing, this article has serious flaws, additional experiments are needed and I am not convinced about correctness of conducted study. 

Reviewer 3 Report

Comments and Suggestions for Authors

In this study, the bioactivity of CYP24A1-resistant 2α-substituted vitamin D derivatives was investigated in vivo and in osteoblasts and DFAT cells. The authors in their study entitled "2α-Substituted Vitamin D Derivatives Effectively Enhance the Osteoblast Differentiation of Dedifferentiated Fat Cells" in vitro investigate the ability of 2α-substituted 1,25(OH)2D3 metabolites to induce VDR activation as well as the induction of VDR downstream target gene expression of CYP24A1 in vivo in vitro. In additional experiments, the authors examined the effect of these 2α-substituted 1,25(OH)2D3 derivatives in the induction of osteoblast differentiation of DFAT cells compared to the active vitamin D hormone. The results showed an induction of intestinal CYP24A1 gene expression by oral administration of 2α-substituted vitamin D derivatives in mice as well as in additional in vitro experiments in osteoblasts, HEK293 and MG-63 cells. In addition, these synthetic vitamin D derivatives induced VDR activation in HEK239 cells and ALP activity in DFAT cells similarly or in some cases more effectively than 1,25(OH)2D3. The results provide evidence for the bioactivity of 2α-substituted vitamin D derivatives in the regulation of the activity of the transcription factor VDR and its downstream gene targets, and a role in the induction of differentiation of DFAT cells into an osteoblastic phenotype like or more effective than the active vitamin D hormone calcitriol.

Nevertheless, the study has some relevant limitations and weaknesses that need to be addressed in a revision prior to publication.

1)     The animal experiment is inadequately described in the Methods section because the description of the organ collection is missing. Furthermore, it is not entirely clear why the animals were sacrificed between 6 and 8 h after oral administration of the compounds. Since mRNA was measured as the outcome of the experiment, and the vitamin D derivatives used are very potent compounds, a 2h variance in mortality between animals may cause a bias in the measured results. Therefore, the question arises whether the reported stronger effect of O2C2 and O2C3 is mediated by a time shift in organ sampling within or between experimental animal groups rather than by themselves. Since 1,25(OH)2D3 also regulates serum phosphorus, this should be measured in an additional experiment to prove the 1,25(OH)2D3-like efficacy of the 2α-substituted compounds. Since the authors describe a variance in the efficacy of the investigated metabolites in in vivo results compared to the in vitro results of CYP24A1 induction, the generated data showing a more effective intestinal CYP24A1 induction mediated by the synthetic 2α derivatives should be reproduced in human CaCo-2 cells. Also, in order to exclude a possible bias due to the heterogeneous sampling of the organs used and a possible effect of this on the shown results.

2)     The reported effect of induction of osteoblastic differentiation in DFAT cells by treatment with 2α-substituted vitamin D compounds is based only on a single experiment measuring the effect on alkaline phosphatase activity after 7 days of treatment as a marker of differentiation. Given that the authors describe a resistance of the 2α compounds used to the degradation enzyme of active vitamin D metabolites, CYP24A1, how can the authors conclude that the 2α compounds are more effective in inducing the activity of ALP when it is obvious that this can be explained simply by the degradation of the included positive control, 1,25(OH)2D3, during a 3-day incubation period without medium change, whereas the 2α compounds remain unaffected by this degradation during the same period. Therefore, the authors would have to demonstrate that the same concentrations of the substances are still present in the medium after 3 days without changing the medium, or they would have to repeat the experiments, changing the medium daily and adding new compounds.

3)     Since the study is titled that 2α-vitamin D derivatives are able to induce osteoblastic differentiation in DFAT and the only experiment to show this is methodologically poorly designed as mentioned in the point above, additional parameters should be measured to confirm the postulated hypothesis. To this end, other targets of an osteoblastic or adipogenic status of DFAT cells after treatment with those vitamin D derivatives should be measured in terms of gene and protein expression. Adipogenic markers such as FABP4, PPARγ, adiponectin or perilipin2 and osteoblastic markers such as osteopontin, osteocalcin, Runx2, BMP2, BSP and Col1A1 should be measured.

4)     Since the other experiments focused exclusively on CYP24A1 as a VDR target gene, but here a regulation towards an osteoblastic phenotype is postulated, the already used MG-63 cells should also be used to identify the proposed osteoblast marker genes and their regulation by the 2α-vitamin D derivatives as a cellular positive control of this cell line by qRTPCR.

5)     It is unclear under section 3.7 RT-PCR, why the relative mRNA levels were calculated relative to those of 1,25(OH)2D3. They should be calculated relative to the EtOh controls, which is also incorrectly stated in the accompanying figure captions. So it was either calculated incorrectly or described incorrectly and needs to be corrected. This is also relevant to the description of the in vivo data, where the mRNA values were calculated relative to those of 1,25 treated mice instead of control mice, which is incomprehensible.

6)     Why is Tuckey's multiple test used for the data in Figure 3? When data are compared only to control values, one-way ANOVA with Dunnett's multiple comparison test should be used. Otherwise, the significance levels of the Tuckey's test between 1,25(OH)2D3 and the 2α-substituted metabolites should be reported because the differences look strong.

7)     The quality of Figures 2, 3, 4, 5 and 6 is inappropriate because of the size, the font size of the x-axis, the colors used and the pattern of the graph, which hide the individual data points.

8)     50% of the citations used are from 2009 or even older and nearly 30% of the citations seem to be from the group itself. And the fact that blood calcium levels in rats are increased by these 2α metabolites is not found in the cited literature (line 59).

9)     The introduction and discussion section are very sparse and focuses only on the groups own data in previous studies. In particular, the in vivo results should be discussed in more detail, as well as the observed differences between the effects of 2α-derivatives in vitro compared to in vivo in the regulation of CYP24A1 compared to 1,25(OH)2D3. The strong focus in the discussion on the relevance of the data with regard to a potential use of the investigated compounds as VDR-targeting drugs in regenerative medicine is incomprehensible in view of the sparse results and somehow not perfectly designed experiments and is overemphasized in the current form.

10)  The concentrations used for the data in Fig. 2 should be given in detail in the caption or in the Methods section.

Comments on the Quality of English Language

11)  Text and spelling errors in lines 64, 134, 207, 271, 278 should be corrected

Round 2

Reviewer 1 Report

Comments and Suggestions for Authors

The revised article has gained in clarity and soundness. The additional experiments implemented the information about the biological activity of all the tested compounds and the conclusions are more consistent with the presented evidence. The article is now ready for publication.

Author Response

Thank you very much.

Reviewer 2 Report

Comments and Suggestions for Authors

Authors have improved their article since the previous submission where it was rejected. After reviewing all the changes made by the authors, I must admit that a current version of their work is filled with valuable information that was previously missing. However, at this time, the article still has several deficiencies that need to be addressed.

Please find my comments below.

1. The authors did not examine the phenotype of isolated DFAT cells prior to their experiments. They used a previously described protocol and trusted its efficacy. In my opinion, this is not a good way to do basic research. Each new batch of primary cells may be different from those previously isolated, even if the protocol is 100% replicated. Cells should at least be cytometrically evaluated to know how heterogeneous the isolated cell population is. Without this knowledge, authors can only assume that they have used DFAT cells in the current experiment. They cannot draw strict conclusions about the properties of DFAT cells without a full evaluation of their phenotype. This kind of research cannot be based on authors' assumptions. It requires hard evidence confirmed by unambiguous data, which the authors did not provide.

2. The relative light units in the caption of Figure 2 have been explained. However, if the data presented is compared to the control, which was taken as 100%, then the data presented is no longer a relative light unit, but a fold change. Please just add an information in brackets to the title of the y-axis: “(% to control)”.

3. Figure 5 shows that O1C2 and O1C3 induce the expression of SPP1 and BGLAP, suggesting that this derivative promotes osteogenesis. The hydroxyalkoxylated derivatives were compared here with the hydroxyalkylated derivatives in one assay. Next, the authors decided to look at ALP expression, but this time the hydroxyalkoxylated derivatives and the hydroxyalkylated derivatives were compared separately, as shown in Figures 6 and 7. Why?

Reviewer 3 Report

Comments and Suggestions for Authors

The changes made by the authors and the addition of new experiments have clearly improved the scientific soundness of the manuscript, allowing it to be published in this form.

Author Response

Thank you very much.

Round 3

Reviewer 2 Report

Comments and Suggestions for Authors

I am satisfied with Authors' answers.